

# Overexpression of *OsFTL10* induces early flowering and improves drought tolerance in *Oryza sativa* L.

Maichun Fang[1,2,3,*], Zejiao Zhou[1,*], Xusheng Zhou[1], Huiyong Yang[1], Meiru Li[2,3] and Hongqing Li[1]

[1] Guangdong Provincial Key Laboratory of Biotechnology for Plant Development, South China Normal University, Guangzhou, People's Republic of China
[2] Key Laboratory of Plant Resources Conservation and Sustainable Utilization, South China Botanical Garden, Chinese Academy of Sciences, Guangzhou, People's Republic of China
[3] Guangdong Provincial Key Laboratory of Applied Botany, South China Botanical Garden, Chinese Academy of Sciences, Guangzhou, People's Republic of China
[*] These authors contributed equally to this work.

## ABSTRACT

Flowering time control is critically important for the reproductive accomplishment of higher plants as floral transition can be affected by both environmental and endogenous signals. Flowering Locus T-like (*FTL*) genes are major genetic determinants of flowering in plants. In rice, 13 *OsFTL* genes have been annotated in the genome and amongst them, *Hd3a* (*OsFTL2*) and *RFT1* (*OsFTL3*) have been studied extensively and their functions are confirmed as central florigens that control rice flowering under short day and long day environment, respectively. In this report, a rice *OsFTL* gene, *OsFTL10*, was characterized, and its function on flowering and abiotic stress was investigated. The expression level of *OsFTL10* was high in young seedlings and shown to be induced by $GA_3$ and drought stress. Overexpression of *OsFTL10* resulted in earlier flowering in rice plants by up to 2 weeks, through up-regulation of the downstream gene *OsMADS15*. *OsFTL10* also regulated *Ehd1* and *OsMADS51* through a feedback mechanism. The OsFTL10 protein was also detected in both nucleus and cytoplasm. Furthermore, yeast two hybrid (Y2H) and bimolecular fluorescence complementation (BiFC) results show that OsFTL10 could interact with multiple 14-3-3s, suggesting that OsFTL10 might function in a similar way to Hd3a in promoting rice flowering by forming a FAC complex with 14-3-3, and OsFD1. Further experiments revealed that constitutive expression of *OsFTL10* improved the drought tolerance of transgenic plants by stimulating the expression of drought responsive genes. These results suggest that rice *FTL* genes might function in flowering promotion and responses to environmental signals.

# INTRODUCTION

Flowering is one of the most important physiological processes that changes the vegetative and reproductive growth of plants. Floral transition can be induced by both environmental factors and endogenous genetic networks. Upon flowering induction, multiple signals

Corresponding authors
Meiru Li, limr@scbg.ac.cn
Hongqing Li, hqli@scnu.edu.cn

converge in floral pathway integrators which subsequently activate downstream genes for growth phase transition and the development of flower organs (*Wagner, 2017*). Recently, advances in the molecular genetics of Arabidopsis and rice, have revealed the molecular nature of florigen to be a globular protein, named FLOWERING LOCUS T (FT), which is produced in the mature leaf and transported to shoot meristem to induce flowering. This process satisfies the major prerequisites of florigen as a systemic floral signal (*Corbesier et al., 2007*; *Tamaki et al., 2007*).

*Hd3a*, the Arabidopsis *FT* homolog protein in rice, belongs to the phosphatidylethanolamine binding protein (PEBP) family in animals (*Kardailsky et al., 1999*). Both *Hd3a* and *FT* are expressed in mature leaves when flowering-promotive day lengths are given, long days (LD) for Arabidopsis and short days (SD) for rice (*Corbesier et al., 2007*; *Tamaki et al., 2007*). Both proteins migrate from leaves to the apical meristem (SAM), where they bind to transcription factors of the bZIP family, including FD in Arabidopsis and OsFD1 in rice. The rice Hd3a and FD1 have been shown to form a florigen activation complex (FAC) with 14-3-3 (GF14 in rice) to promote floral initiation (*Tamaki et al., 2007*; *Taoka et al., 2011*).

Genome annotation results have revealed thirteen FT homologs (*OsFT-LIKE* genes) in the rice genome (*Chardon & Damerval, 2005*). Amongst the thirteen FTLs, *RFT1/FT-L3* showed the highest homology with Hd3a, and was recently found to be important for rice flowering under LD conditions. *FTL/FT-L1* is another member with high homology to Hd3a. Transgenic rice plants overexpressing *RFT1* or *FTL* exhibited an early flowering phenotype, similar to *Hd3a*-overexpressing plants (*Izawa et al., 2002*; *Kojima et al., 2002*). Knockdown of *Hd3a* or *RFT1* in the japonica rice cultivar Norin 8, delays rice flowering only under SDs or LDs, respectively (*Komiya et al., 2008*; *Komiya, Yokoi & Shimamoto, 2009*). In contrast, flowering in rice is almost completely blocked even under favorable SD conditions by knockdown of both *Hd3a* and *RFT1*, suggesting that these two genes have overlapping functions and are necessary for flowering (*Komiya et al., 2008*). Until now, studies in rice have revealed that the photoperiod is a major environmental factor influencing rice flowering time. Furthermore, photoperiodic flowering in rice is considered as two distinct pathways: one including OsGI (an orthologue of *Gigantea*), Hd1 (Heading date 1) and Hd3a. These proteins form a pathway of OsGI-Hd1-Hd3a, which is similar to that of Arabidopsis under SD conditions (*Turck, Fornara & Coupland, 2008*; *Deng et al., 2017*). Another one including Ghd7 (Grain Number, Plant Height and Heading Date 7), Ehd1 (Early heading date 1) and Hd3a/RFT1.These proteins form a pathway of Ghd7-Ehd1-Hd3a/RFT1 in rice under LD conditions. The major genes downstream of Hd3a and RFT1 are *OsMADS14*, *OsMADS15*, *OsMADS18* and *OsMADS34*. Simultaneous knockdown of the above four genes completely blocked flowering in rice (*Kobayashi et al., 2012*; *Tsuji, 2017*).

Floral transition can be induced by a variety of external factors, including various environmental conditions and hormone treatments, such as the well-studied photoperiod pathway in rice and Arabidopsis, as well as the vernalization and $GA_3$ (Gibberellic Acid) pathway in Arabidopsis (*Kazan & Lyons, 2016*). Drought stress influences plant growth and crop yield in many areas of the world, and has been shown to significantly influence

flowering time. During drought stress, plants may adopt the strategy of 'drought escape' by speeding up or delay of flowering (*Bernal, Estiarte & Peñuelas, 2011*; *Franks, 2011*), or the strategy of 'drought avoidance' or 'drought tolerance' by reducing water loss to prevent dehydration (*Kooyers, 2015*). In Arabidopsis, drought stress affects flowering time through transcriptional regulation of FT and TWINSISTER OF FT (TSF). Under long day conditions, induction of the expression of FT and TSF by drought stress depends on GI and abscisic acid (ABA), whilst under short days, floral repressors are induced by ABA and drought, and the transcription of TSF and FT is inhibited (*Riboni et al., 2013*). The Grain Number, Plant Height and Heading Date 7 (Ghd7), which delays heading and increases yield in rice under long days (*Xue et al., 2008*), also affects drought tolerance in rice (*Weng et al., 2014*; *Kazan & Lyons, 2016*).

In rice, the function of *FTL* genes on the induction of flowering has been well-characterized based on the studies of *Hd3a* and *RFT1*. The precise involvement of florigens in environmental response is largely unknown. When investigating the response of rice *FTLs* to hormones and abiotic stresses, we found that only *OsFTL10* expression was stimulated by mannitol and ABA. In this report, we studied the expression of *OsFTL10* and found that it expressed at a higher level in young leaf, and also its expression can be induced by drought stress. Overexpression of *OsFTL10* can induce early flowering in rice through a similar mechanism as that of *Hd3a*. We also found that overexpression of *OsFTL10* can enhance the drought tolerance of plants.

## MATERIALS & METHODS

### Vector construction and rice transformation

To obtain the OsFTL10 overexpression construct, a fragment containing the complete OsFTL10 cDNA (Os05g0518000) was amplified with the primer set Fw-OsFTL10-EcoRI/Re-OsFTL10-EcoRI (Table S1), and cloned into the EcoRI site in binary expression vector pCAMBIA1390. To make the RNAi construct, the cDNA fragment was amplified with the primer set Fw-OsFTL10-RNAi/ Re-OsFTL10-RNAi, and cloned into the pTCK303 vector (*Wang et al., 2004*). To examine the promoter activity in rice, the OsFTL10 promoter (pFTL10), spanning about 2,000 bp upstream of the ATG start codon of OsFTL10 was amplified using PCR. The CaMV35S promoter for β-glucuronidase (GUS) expression in pCAMBIA1301 was replaced with pFTL10. For subcellular localization studies of OsFTL10, its cDNA fragment was amplified by PCR and fused with EGFP in the binary vector pCAMBIA1390. The binary vector for expression of *Hd3a* was from a previous work (*Li et al., 2011a*; *Li et al., 2011b*). The above constructs were introduced into Agrobacterium tumefaciens strain EHA105 by electroporation (*Li et al., 2011a*; *Li et al., 2011b*). Agrobacterium-mediated transformation of rice (Oryza sativa L. ssp. japonica., Zhonghua 11) was performed as described (*Li et al., 2011a*; *Li et al., 2011b*).

### Plant materials and growth conditions

Mature seeds of Zhonghua 11 (Oryza sativa L. ssp. japonica.) were collected from WT and homozygous T2 transgenic plants. Seeds were germinated for 2 days at 28 °C in the

dark, after which they were planted in soil and grown under greenhouse conditions to maturity (12-h light at 30 °C/12-h dark at 22 °C). Mature leaves from 30 day old plants were collected at 10 O'clock in the morning, and RNAs were extracted for qRT-PCR analysis of the expression of flowering related genes. The major agronomic traits including heading date, plant height, grain weight and number of flowers per panicle of the homozygous plants were investigated.

## FTL10 sub-cellular localization and BiFC analysis

The subcellular localization of FTL10 and GF14c, as well as BiFC analysis characterizing the interaction of FTL10 and GF14c, were carried out according to (*Lin et al., 2017*). For subcellular localization studies of FTL10 and GF14c, the coding regions of both FTL10 and GF14c were amplified with primers (Table S1) and fused with EGFP, respectively. For BiFC analysis, *GF14c* and *OsFTL10* were amplified by PCR and restriction-cloned (*Bam* HI and/or *Eco* RI) into the p35S::cYFP and p35S::nYFP vectors, respectively. Negative controls constructs were made by replacing GF14c or OsFTL10 with GUS (*Taoka et al., 2011*). The above vectors were transformed into rice protoplasts for live cell imaging. Images were captured using a LSM 710 confocal microscope (Zeiss, Jena, Germany) (*Lin et al., 2017*).

## Yeast two-hybrid assays

Yeast two-hybrid (Y2H) assays were performed using the Matchmaker$^{TM}$ Gold Yeast Two-Hybrid Systems (Clontech, Mountain View, CA, USA) according to *Lin et al. (2017)*. The open reading frames of *OsFTL10* and 14-3-3s were inserted into pGBKT7 and pGADT7 vectors with proper restriction enzyme sites indicated in the primers to create bait and prey, respectively.

## GUS staining of transgenic rice plants

Homozygous transgenic lines carrying *the pFTL10*::*GUS* were randomly chosen for the study. Seeds were briefly sterilized and germinated, then planted in soil and grown under greenhouse conditions. Immature and mature leaf, roots and other tissues were collected from the plants grown at different stages, GUS staining of the plant materials was performed according to procedures described by *Jefferson (1987)*.

## Growth regulators and abiotic stress treatments

WT seedlings germinated for 2 weeks on half strength MS medium were collected and incubated in ddH$_2$O containing either of the following components: 5 mmol/L IAA, 1 mmol/L 6-BA, 0.1 mmol/L MeJA, 10 mmol/L GA$_3$, 100 µmol/L ABA, 1 mmol/L SA, 200 mmol/L Mannitol, 200 mmol/L NaCl, respectively. Treatments with IAA and BA were performed for 0, 0.5, 1 and 4 h, respectively. Treatments with ABA, SA, MeJA and GA were performed for 0, 1, 4 and 12 h, respectively. Treatments with mannitol and NaCl were performed for 0, 6, 14 and 48 h, respectively. All the treatments were started from about 10 O'clock in the morning. After treatment for the indicated times, total RNAs were extracted and reversely transcribed as the template for qRT-PCR.

## qRT-PCR for gene expression analysis

Total RNAs were extracted using a TRIzol kit according to the user's manual (Invitrogen, Carlsbad, CA, USA). For each sample, 1 μg of total RNAs was reverse-transcribed as cDNA template using a kit (TaKaRa, Kusatsu, Japan). qRT-PCR was carried out in a mixture of 20 μL consisting of cDNA, gene specific primers (Table S1) and 10 μL 2X mix (TaKaRa), water was added to the mixture to a final of 20 ul. qPCR was performed on an ABI 7900 real-time PCR machine according to the manufacturer's instruction (Applied Biosystems). For each sample, three biological replicates were performed and cDNA were amplified in triplicate by quantitative PCR. The relative expression values were determined by using rice *Ubiquitin* gene as reference and the comparative Ct method (2- $\Delta\Delta$Ct).

## Drought stress treatment

Drought stress treatment of WT and transgenic plants was performed according to *Li et al. (2011a)*; *Li et al. (2011b)*. Seeds of WT, overexpression transgenic lines ox2, ox5, and ox8 and RNAi lines i1,i2 and i5 (T3, homozygous seeds) were germinated, and in the same pot containing vermiculite were soaked with distilled water. Eight days later, water was withheld from the seedlings. After another 20 (for RNAi lines) or 25 days (for overexpression lines), the plants were re-watered for recovery. Three days later, plants with green, healthy leaves were regarded as viable, having survived. For study of drought related gene expression in transgenic plants, seedlings germinated for 2 weeks were incubated in a solution containing 200 mmol/L mannitol for 2, 4, 8, and 24 h, respectively. Total RNA was extracted for qRT-PCR.

# RESULTS

## Expression pattern of *OsFTL10* in rice

*OsFTL10* encodes a protein consisting of 174 amino acid residues. To investigate the temporal and spatial gene expression pattern of *OsFTL10* in rice, we quantified the *OsFTL10* expression level in different tissues including immature and mature leaves, stem, young panicle and floral organs from the wild-type plants grown under SD conditions. *OsFTL10* expression was detected in all the above tissues. The highest expression level was observed in immature leaves with a lower level of expression found in floral organs (Fig. 1A).

The expression pattern of *OsFTL10* was also investigated by GUS staining assay in stable transgenic plants. The promoter region of *OsFTL10* was cloned and a construct was made by placing the *GUS* reporter under the *OsFTL10* promoter. The construct was transformed into rice cell and stable transgenic plants were obtained. *GUS* expression was found in immature leaf, mature leaf, flower, young panicle, pollen and shoot tip (Figs. 1B–1H). Strong GUS staining was found in immature leaf tissue and very faint blue staining was found in the pollen (Fig. 1G).

## Expression of *OsFTL10* under growth regulators and abiotic stress treatments

To determine whether *OsFTL10* expression is influenced by environmental factors, qRT-PCR was performed to study the expression of *OsFTL10* under different growth regulators

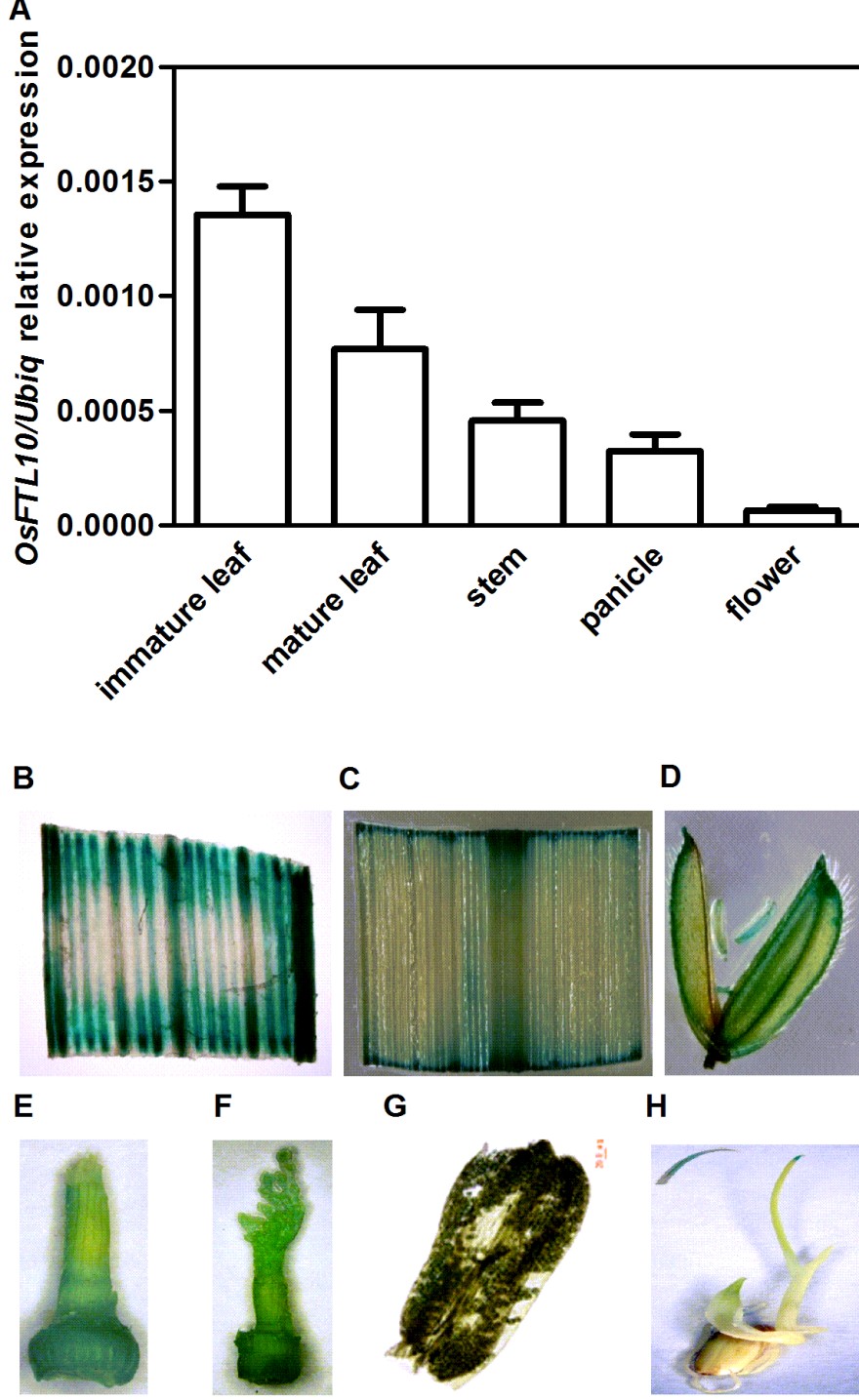

**Figure 1** **Temporal and spatial expression *OsFTL10* in rice.** (A) The expression of *OsFTL10* in rice tissues by qRT-PCR analysis. (B) GUS assay of OsFTL10 promoter activity in immature leaf. (C) Mature leaf. (D) Flower. (E, F) Young panicle. (G) Anther. (H) Germinated seedling (with amplified leaf tip).

and abiotic stress treatments. Seedlings germinated for 2 weeks on half strength MS medium were collected and treated with different growth regulators and abiotic stresses for different time intervals, and total RNAs were extracted for qRT-PCR. Amongst the treatments of different growth regulators, ABA at 100 μM stimulated the expression of *OsFTL10* to nearly two-fold after 4 hrs (Fig. 2C). The addition of GA₃ at 10 mM greatly increased the expression of *OsFTL10* to more than ten-fold its basal level (Fig. 2F). Treatments with SA and MeJA also showed slightly increased expression of *OsFTL10* (Fig. 2D, 2E), whilst treatment with IAA or BA reduced the expression level of *OsFTL10* (Figs. 2A, 2B). Abiotic stress treatment was performed using 200 mM mannitol or 200 mM NaCl. Treatment with 200 mM mannitol resulted in increased *OsFTL10* expression to four-fold basal levels at 6 hrs, and the expression dropped after 24 h (Fig. 2G). 200 mM NaCl slightly increased the expression of *OsFTL10* which dropped after 24 h (Fig. 2H). In accordance with the above results, some *cis* elements involved in drought stress and ABA response were found in *OsFTL10* promoter (Table S1). As ABA and mannitol are related to drought stress, we further studied the drought tolerance of *OsFTL10* transgenic plants in addition to its function in flowering.

## Overexpression of *FTL10* induced earlier flowering in rice

As *OsFTL10* is one of the homologs of *Hd3a*, we expected that overexpressing this gene would also promote rice flowering. We generated *OsFTL10* overexpression transgenic plants and examined the transgene expression by RT-PCR (Fig. 3A). At the same time, we obtained RNAi transgenic plants for *OsFTL10*, and in some lines *OsFTL10* expression was greatly reduced (Fig. 3B). Representative pictures of the overexpressing line ox8 (Fig. 3A) and the RNAi line (Fig. 3B) are shown. To investigate the agronomic traits of the transgenic plants, the homozygous T3 generation of the three independent overexpressing lines ox2, ox5, ox8 and RNAi lines i1, i2, i5 were grown under short day conditions, and their phenotype data compared with WT plants (Fig. 3). The average heading date of the overexpressing lines was 53 days, which was about 2 weeks less than of the WT plants (about 68 days). The RNAi lines did not show significant difference in heading time compared with the WT (Fig. 3C). The plant height for the overexpressing lines was on average 65 cm (Fig. 3D), which was about 19 cm shorter than that for the WT (84 cm). The number of grains per panicle and grain weight were also reduced significantly in the overexpressing lines, compared to that in WT (Figs. 3E, 3F). In the RNAi lines, the above traits were not significantly changed (Figs. 3C–3F). These results showed that overexpression of *OsFTL10* could induce early flowering, which influenced the agronomic traits of the plants. The down-regulation of endogenous *OsFTL10* exerted little effect on plant growth. When compared with *Hd3a* overexpression plants, we found that heading time of *OsFTL10* overexpression plants were about 3 weeks longer (Table S1).

To elucidate the function of *OsFTL10* on flowering induction, we first examined the subcellular localization of the OsFTL10 and GF14c protein. Constructs were made to fuse the OsFTL10 or GF14c with EGFP protein, under the CaMV35S promoter (*CaMV35S::OsFTL10-EGFP*). The constructs were transformed into rice leaf mesophyll protoplasts, and the EGFP signal was analyzed. The EGFP signal of OsFTL10 was observed

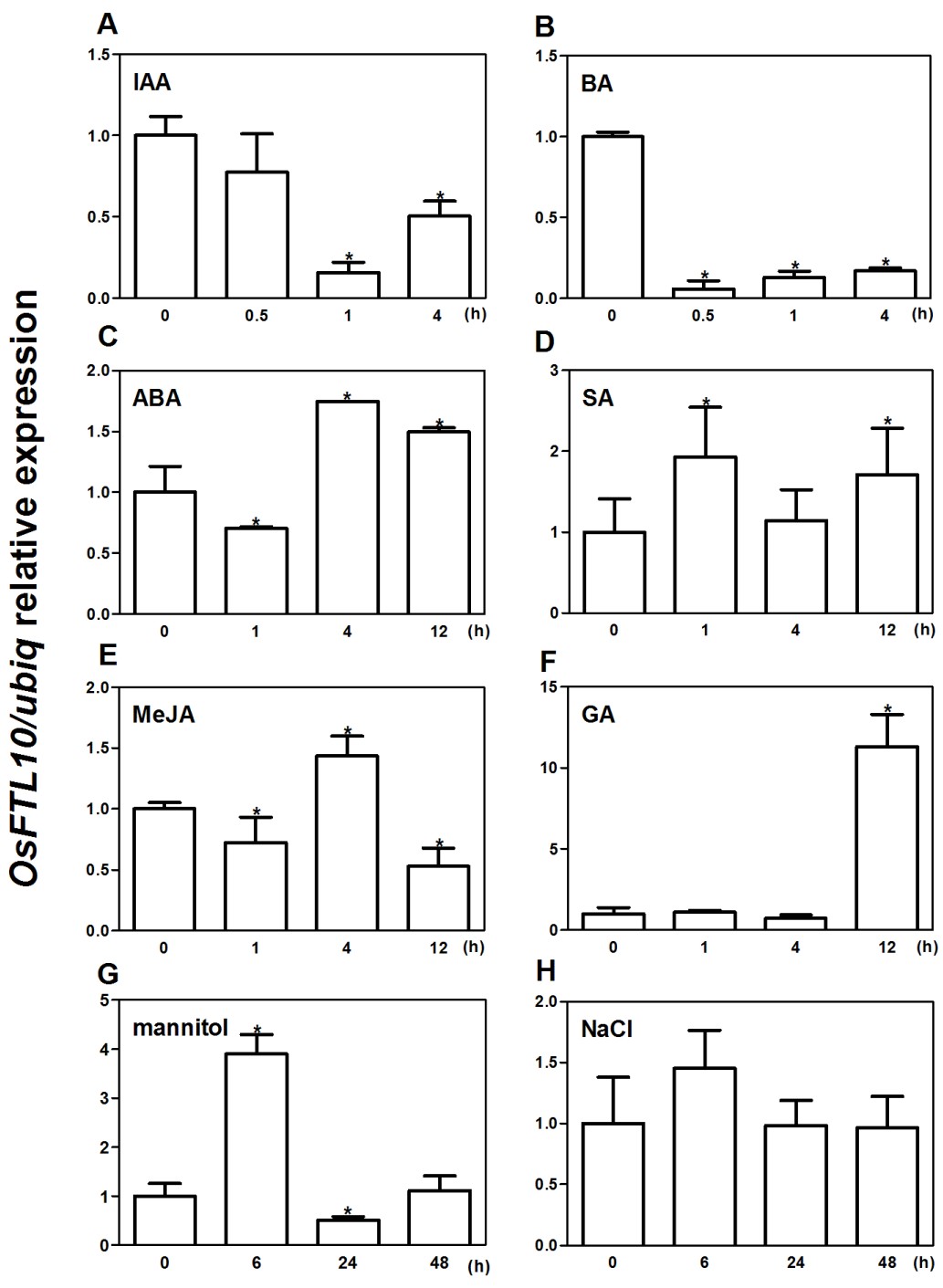

**Figure 2** **Expression of *OsFTL10* in rice seedlings treated with different growth regulators and stresses.** Expression of *OsFTL10* in rice seedlings treated with IAA (A). (B) BA. (C) ABA. (D) SA. (E) MeJA. (F) GA. (G) Mannitol. (H) NaCl. All data represent the mean of three biological replicates, with error bars indicating SD. (* indicates a significant difference in comparison with the control (0 h) at $P < 0.05$).

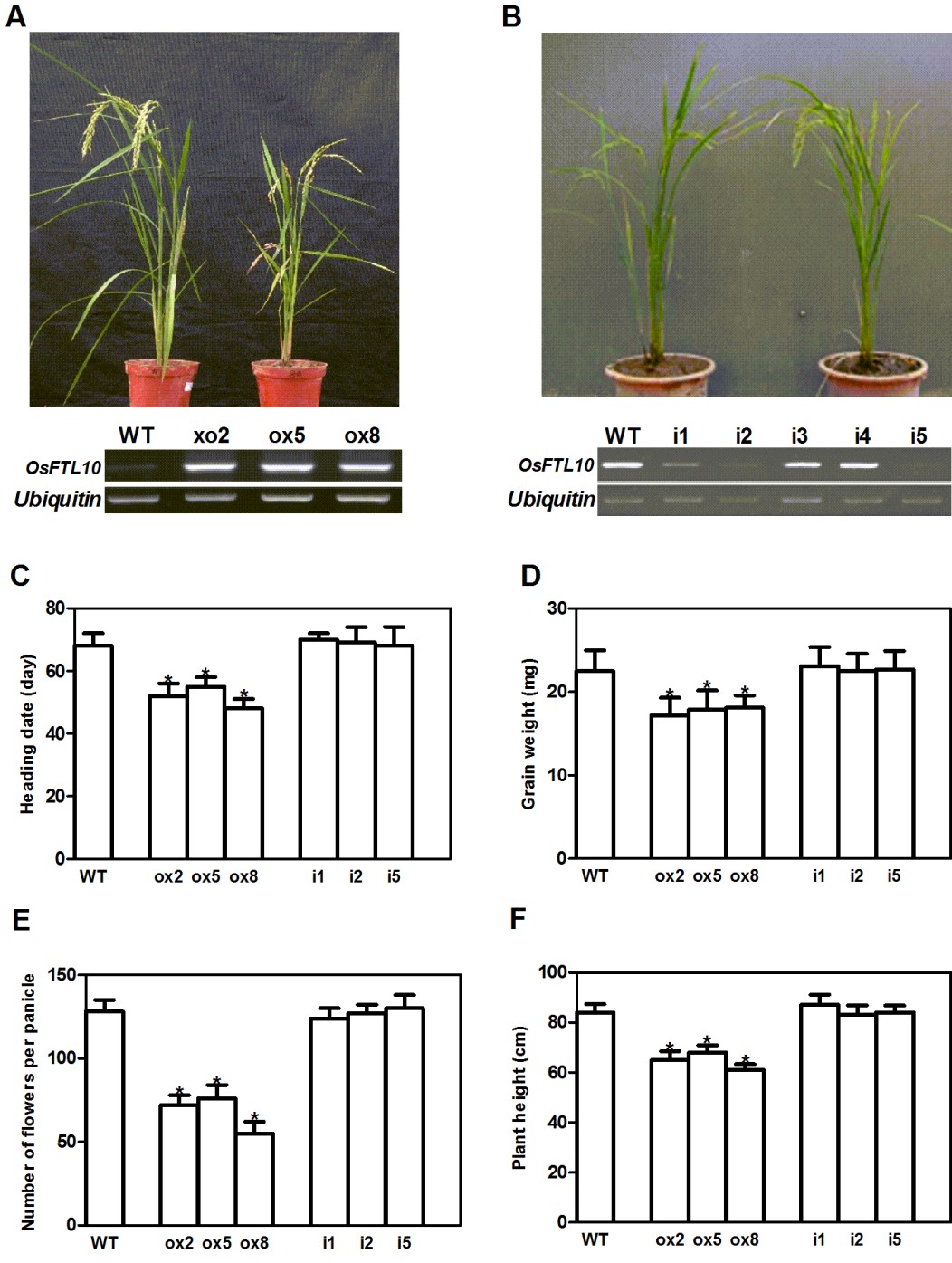

**Figure 3** **Phenotype of *FTL10* overexpression and RNAi transgenic plants.** (A) Representative picture of WT and overexpression line ox8 after heading. (B) Representative picture of WT and RNAi line i5 after heading. Performance of the transgenic lines in greenhouse on heading date (C). Grain weight (D). Number of flowers (E). Plant height (F). Data represent the mean of 15 to 24 plants, with error bars indicating SD. (* indicates a significant difference in comparison to the WT at $P < 0.05$.)

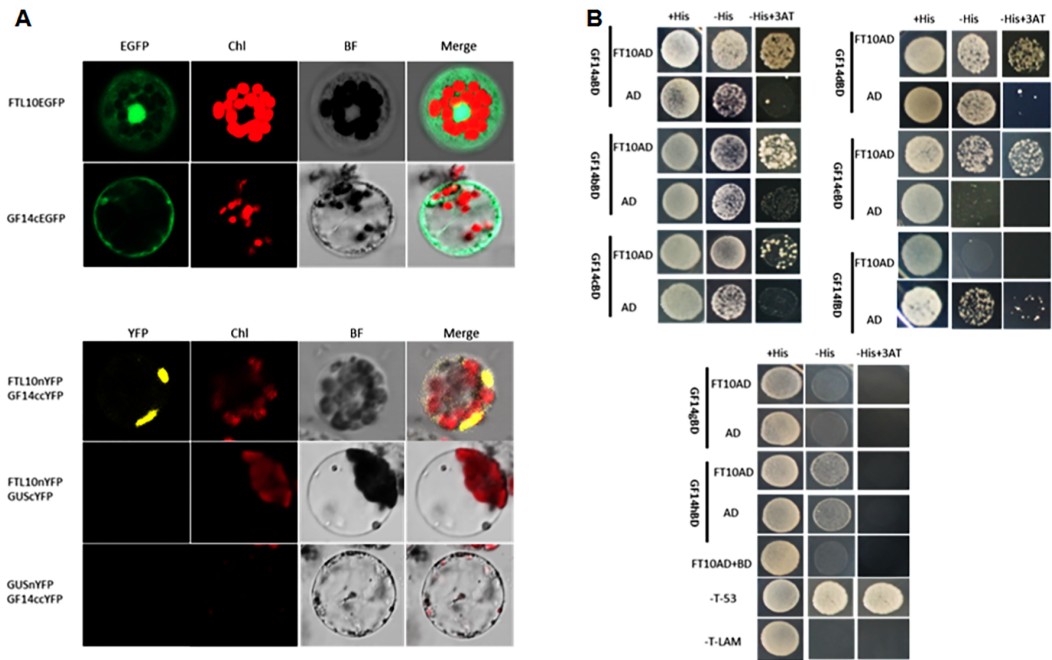

**Figure 4** **BiFC and Y2H analysis of OsFTL10 interaction with 14-3-3c.** (A) Subcellular localization of OsFTL10, 14-3-3c. (B) BiFC analysis of OsFTL10 interaction with 14-3-3c in rice protoplasts. (C) Y2H analysis of interaction of OsFTL10 with different 14-3-3s.

in both the nucleus and cytoplasm, whilst the EGFP signal of GF14c was observed mainly in the cytoplasm (Fig. 4A). These results indicated that FTL10 had a similar subcellular localization to Hd3a.

Hd3a was found to interact with 14-3-3, and form a complex with OsFD1 in the nucleus for activation of the downstream flowering genes (*Taoka et al., 2011*). The interaction of OsFTL10 with GF14c in the rice protoplast was examined by BiFC. A strong signal was detected in the nucleus, which demonstrated the interaction of the two proteins. At the same time, the negative controls showed no obvious signal (Fig. 4B). The interactions of OsFT10 with multiple 14-3-3 s were also investigated by the yeast two-hybrid system. The results showed that OsFTL10 interacted with different 14-3-3 homologs in rice, including GF14a, GF14b, GF14c, GF14d, GF14e (Fig. 4C). These results suggested that OsFTL10 functioned similarly to Hd3a to bind with 14-3-3 s, which was a key step for entering the nucleus in the formation of the FAC complex with OsFD1 (*Taoka et al., 2011*).

Flowering in rice is mainly influenced by the photoperiodic pathway (*Tsuji, Taoka & Shimamoto, 2011*). *OsGI* (an orthologue of *Gigantea*) is an integrator of photoperiod pathway in rice (*Kim et al., 2008*). Under short day (SD) conditions, *OsGI* up-regulates the expression of *Hd1* (a CONSTANS-like gene), and *Hd1* will induce the expression *Ghd8* (*grain yield, heading date and plant height 8*). *Ghd8* encodes the *OsHAP3* subunit of a CCAAT-box binding protein and can activate the expression *of Hd3a* (*Yan et al., 2011*). *Ehd1* (*Early heading date 1*) is a B type response regulator, which up-regulates the expression of FT-like genes (*Matsubara et al., 2011*). *OsGI* can also up-regulate *Ehd1*

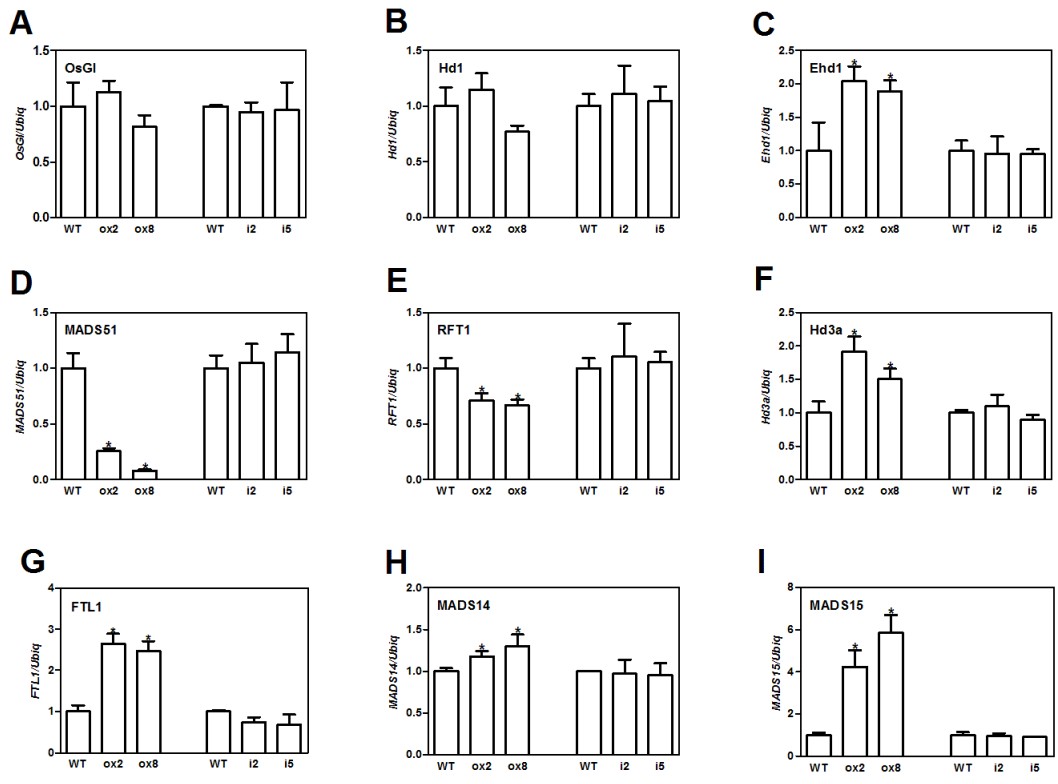

**Figure 5** **Expression of major flowering related genes in transgenic plants.** Expression of OsGI (A), Hd1 (B), Ehd1 (C), MADS51 (D), RFT1 (E), Hd3a (F), FTL1 (G), MADS14 (H), MADS15 (I) in transgenic plants. All data represent the mean of three biological replicates, with error bars indicating SD. (* indicates a significant difference in comparison to the WT at $P < 0.05$).

through activating *MADS51* under SD conditions (*Kim et al., 2008*). To dissect the genetic networks of *OsFTL10* on flowering, we performed qRT-PCR analysis of the genes that played important roles in rice flowering. The cDNA template was obtained from RNA extracted from mature leaves of WT, overexpressing lines ox2 and ox8, and the RNAi lines i2 and i5 before heading. For genes upstream of *Hd3a* and *RFT1*, *GI*, *Hd1*, *Ehd1* and *MADS51* were selected. qRT-PCR results showed that overexpression of *FTL10* had no significant influence on the expression of *GI* and *Hd1* in both ox2 and ox8, as their expression levels were similar to those in WT (Figs. 5A and 5B). *Ehd1* was significantly up-regulated in both overexpressing lines at 1.5 and 2 fold (Fig. 5C). However, *MADS51* expression was largely suppressed in the overexpressing lines (Fig. 5D). Amongst the *FTL* genes, *Hd3a*, and *FTL1* were found to be up-regulated in both ox2 and ox8 (Figs. 5F, 5G), whilst the expression of *RFT1* slightly decreased (Fig. 5E). For the major genes downstream of *Hd3a* (*Taoka et al., 2011*), we found that *MADS15* expression was significantly increased (4 to 6 fold) in ox2 and ox8 (Fig. 5I), and *MADS14* was slightly changed in these two lines (Fig. 5H). The results from the RNAi lines i2 and i5, showed similar results to those in the WT line (Fig. 5). These results suggested that overexpression of *OsFTL10* induced early

flowering, through direct enhancement of downstream *MADS15* expression and feedback regulation of its upstream genes.

## Overexpression of *FTL10* improved drought tolerance in rice

As *OsFTL10* expression can be influenced by different growth regulators and abiotic stresses such as ABA and mannitol treatments, the drought stress tolerance in transgenic plants was investigated. Transgenic and WT seedlings grown under normal conditions with similar vitality were used for treatment (Figs. 6A, 6D). When water was withheld for 25 days from the seedlings of the *OsFTL10* overexpressing plants (ox2, ox5, and ox8) and the WT plants, obvious drought stress phenotypes were observed (Fig. 6B). Compared with the WT plants, leaf rolling and browning were delayed in the *OsFTL10* overexpressing plants. Three days after re-watering, most of the *OsFTL10* overexpressing plants survived, whereas the WT plants withered and died (Fig. 6C). The average survival rate for WT, ox2, ox5 and ox8 plants were 6%, 84%, 88% and 91%, respectively (Fig. 6G). Similar treatment was performed on *OsFTL10* RNAi lines (i1, i2, and i5) and the WT plants, except that water withholding time was reduced to 20 days (Fig. 6E). The *OsFTL10* RNAi plants were found to be more sensitive to drought stress, and after re-watering, most of the WT remained viable, whereas the RNAi plants died (Fig. 6F). The survival rate for WT, i1, i2 and i5 plants were 92%, 14%, 6% and 8%, respectively (Fig. 6H). These results suggested correlation of drought tolerance with *OsFTL10* expression. In transgenic rice plants overexpressing *OsFTL10* acquired significantly drought tolerance, whilst the suppression of *OsFTL10* resulted in drought sensitivity.

## Influence of *OsFTL10* on the expression of drought stress-related genes

Environmental stresses and other external signals can induce complex responses in plants including changes in gene transcription, protein synthesis and metabolism. Studies in different plant species have revealed key genes and proteins involved in stress signaling and transcriptional regulation (*Ito et al., 2006*; *Shinozaki & Yamaguchi-Shinozaki, 2007*; *Molina et al., 2008*; *Nakashima et al., 2011*; *Rai & Penna, 2013*; *Janiak, Kwaśniewski & Szarejko, 2016*). To elucidate the mechanisms of *OsFTL10* on drought tolerance, the expression of stress-related transcription factors in stable transgenic plants under drought stress was revealed. Seedlings from WT, i5 and ox8 were collected for drought stress treatment, and RNAs were extracted for qRT-PCR analysis.

*OsDREB2A* is a drought inducible transcription factor found in rice, which shows increased expression upon drought stress treatment and constitutive expression of this gene results in drought tolerance of transgenic plants (*Ito et al., 2006*). The expression of this gene was rapidly induced in WT plants and ox8 upon drought stress, a response which continued for 24 h. However, the expression level increased significantly after 8 h of treatment in ox8 compared to WT plants. In i5, the expression of *OsDREB2A* decreased after 8 h of treatment (Fig. 6I).

bZIP23, a transcription factor that confers ABA-dependent drought resistance in rice (*Xiang et al., 2008*), has also been shown to be induced by drought stress. Overexpression

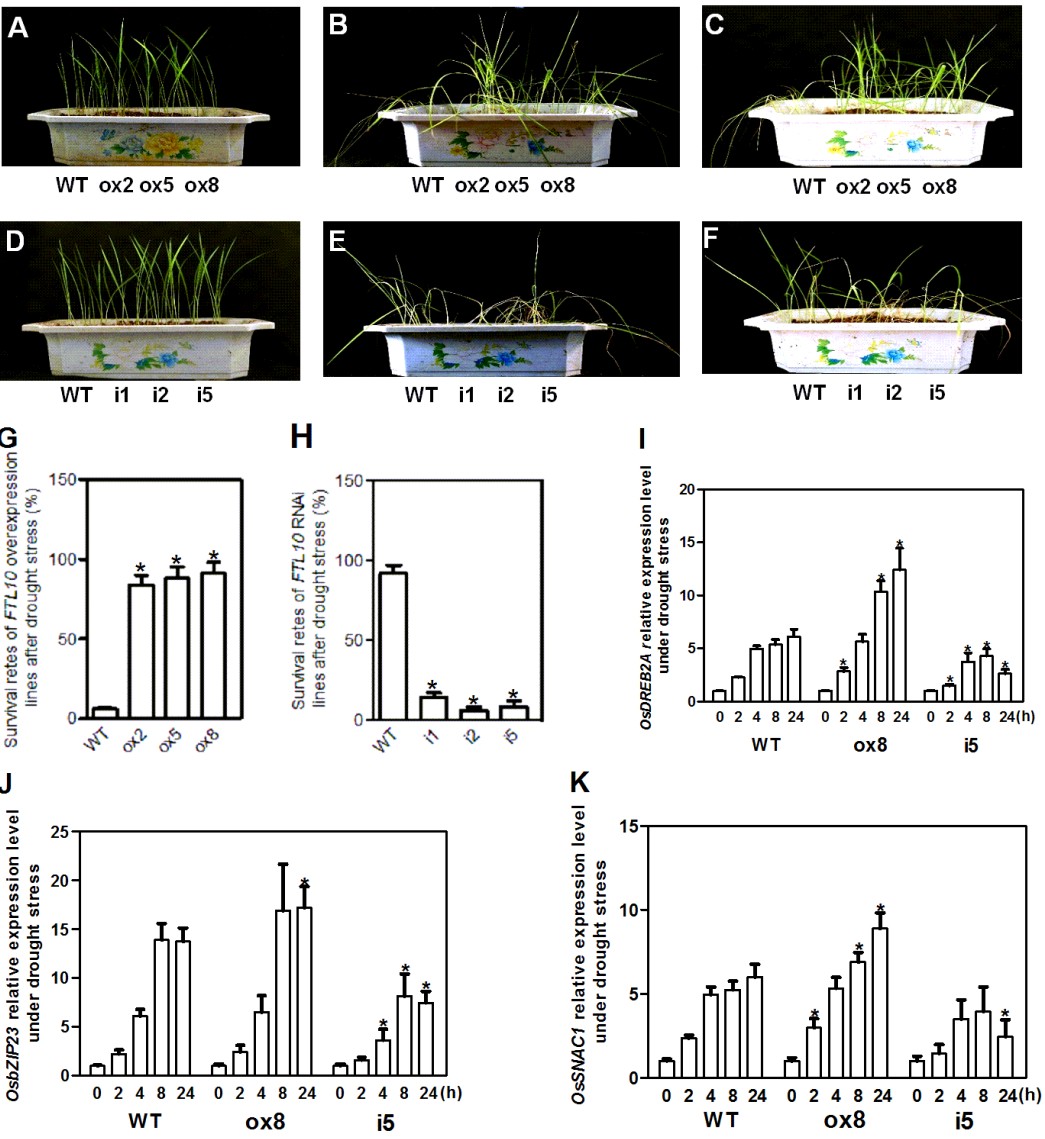

**Figure 6** **Representative picture of drought tolerance test of transgenic plants and expression of drought stress related genes in transgenic plants.** Phenotypic comparison of the transgenic rice plants before (A, D), and after water stress (B, E), and after recovery for 3 days (C, F). (G, H) Survival rates of transgenic plants after drought stress treatment. Data represent the mean of three replicates, with error bars indicating SD. (* indicates a significant difference in comparison to the WT at $P < 0.05$). Expression of stress related genes OsDREB2A (I), bZIP23 (J), and SNAC1 (K) in WT and transgenic plants after drought treatment for different time. Data represent the mean of three replicates, with error bars indicating SD. (* indicates a significant difference in comparison to the WT with the same treatment time at $P < 0.05$).

of this gene resulted in drought tolerance in transgenic plants. The expression of bZIP23 was rapidly induced in WT, ox8 and i5 plants under drought stress, whilst its expression decreased after 8 h in both WT and i5 plants. In ox8, the high expression level of this gene was maintained for 24 h (Fig. 6J).

*SNAC1* gene belongs to the stress-related NAC superfamily of transcription factors. *SNAC1* expression was induced rapidly in WT, i5 and ox8 plants. In ox8, the expression of this gene showed higher expression levels after induction which were maintained after 24 h. Together, these results suggested that overexpression of *OsFTL10* improved drought stress tolerance in rice by modulating the expression of stress responsive genes (Fig. 6K).

## DISCUSSION

In this report, we studied the function of OsFTL10 and found that overexpression of this gene induced earlier flowering in rice. These data are similar to the well-studied rice florigen Hd3a and RFT1, which promoted rice flowering under short and long day conditions, respectively. We also found that overexpression of *OsFTL10* enhanced the tolerance of rice plants to drought stress.

In rice, 13 FT like genes have been found based on sequence similarity, but only three genes, *OsFTL1*, *Hd3a* and *RFT1,* have been studied. Amongst these genes, Hd3a and RFT1 have been extensively studied and shown to be responsive for rice flowering under short and long day conditions, respectively (*Komiya et al., 2008*). Studies have revealed that Hd3a proteins are produced in mature leaves, and then transported to the shoot meristem. Hd3a protein binds 14-3-3 in the cytoplasm, and is imported to nucleus to form a complex (FAC), which binds to C box of *MADS15* promoter region, to stimulate the expression of *MADS15* and initiate flower transition (*Taoka et al., 2011*). To determine whether OsFTL10 also formed a FAC complex to stimulate the floral transition, BiFC was performed and showed that OsFTL10 interacted with GF4c in the nucleus. Yeast two-hybrid results also showed that OsFTL10 interacted with major 14-3-3 homologs in rice, including GIF14b and GIF14c, two 14-3-3 proteins interacting with Hd3a (Fig. 4). We also found that OsFTL10 up-regulated the downstream flowering gene *OsMADS15* (Fig. 5). These data suggested that OsFTL10 had a function similar to Hd3a.

As a homolog of Hd3a, overexpression of *OsFTL10* induced early flowering in rice, but its flowering promotion effect was weaker than Hd3a. The number of days to heading in *CaMV35S::Hd3a* plants was less than 30 days (Supplemental File 1) compared to 53 days in *CaMV35S::OsFTL10* plants. As overexpression of both genes under the same promoter resulted in a different flowering time, this effect is probably caused by the amino acids composition of the two proteins. In addition, we found some differences between the two FTLs: (1) *OsFTL10* was found to have a high expression level in young leaves (Fig. 1A), whilst Hd3a was normally highly expressed in mature leaves; (2) although the localization of both OsFTL10 and Hd3a proteins was found in both the cytoplasm and nucleus (Fig. 4), our BiFC results showed the interaction of OsFTL10 and GF14c mainly occurred in the nucleus, whilst Hd3a and GF14b interactions took place in the cytoplasm (*Taoka et al., 2011*); (3) overexpression of *OsFT10* enhanced the expression of *Ehd1* (Fig. 5), *Ehd1* could activate the expression of *FTLs*, which explains why the expressions of other *FTLs* were influenced. We also found that the expression of *OsMADS51* was suppressed in *OsFTL10* transgenic plants (Fig. 5). Thus, the function of OsFTL10 may be slightly different from the major florigens Hd3a and RFT1 in promotion of flowering.

*Ehd1* can promote rice flowering under both SD and LD. Under SD, *Ehd1* is involved in two modules *OsGI-OsCO-Ehd1-Hd3a* and *OsGI-OsMADS15-Ehd1-Hd3a*, in which *Ehd1* functions upstream of *Hd3a* and promotes flowering by directly up-regulating *Hd3a* expression (*Yan et al., 2011*). The expression of *OsFTL10*, together with the increased expression of *Ehd1* and *Hd3a* in overexpression lines may lead to earlier flowering phenotype in the transgenic plants. As *Ehd1* is a central player which integrates flowering signaling from both SDs and LDs, and is controlled by multiple genes, the feedback regulation of *Ehd1* by OsFTL10 needs further investigation. *OsMADS51* is a flowering promoter under SDs, which is downstream of *OsGI* (Kim et al. 2008). The decreased expression level of this gene in *OsFTL10* overexpression lines is somehow in contradictory with their early flowering phenotype. However, the effects of *OsMADS51* may be reduced by the increased expression of *Ehd1* downstream of it. We also found that *RFT1*, the florigen responsible for flowering under LDs, are slightly down-regulated. We suspect that the expression of this gene was influenced by *OsFTL10* or other flowering related genes, as similar effects had been observed between *Hd3a* and *RFT1*.When *Hd3a* expression was knocked down by RNAi, *RFT1* was activated for promoting flowering under SDs (*Komiya, Yokoi & Shimamoto, 2009*).

Photoperiod and vernalization are two environmental factors which influence flowering time in many plant species (*Khan, Ai & Zhang, 2014*). In the model plant Arabidopsis, several photoperiod and circadian clock genes have been found to affect the expression of drought stress related genes. Mutations in clock components, such as the triple mutant prr5 (PSEUDO-RESPONSE REGULATOR 5), 7, 9 are more tolerant to high salinity and drought stresses. *TOC1* (timing of CAB expression 1) RNAi plants showed improved drought tolerance. *GI* overexpression plants show enhanced salt sensitivity, whereas gi mutants show enhanced salt tolerance and enhanced survival under drought stress and oxidative stress (*Grundy, Stoker & Carre, 2015*). *Phytochrome B (PhyB)* is also a negative regulators of dehydration response (*Legnaioli, Cuevas & Mas, 2009*). In barley, variation at the photoperiod response and clock genes Ppd-H1 and HvELF3 can affect the expression level of osmotic stress genes (*Habte et al., 2014*). Stresses such as drought and salt, can also regulate flowering times (*Kazan & Lyons, 2016*). Early flowering can be induced by drought stress under LD conditions, and ABA can up-regulate the expression of major flower integrators *FT/TSF* and *SOC1* in a photoperiod-dependent manner (*Riboni et al., 2013*). Interestingly, most of the above flowering related genes were found to be negative regulators of stress responses. *OsGI* was recently found to be a negative regulator of osmotic stress response in rice (*Li et al., 2016*). As *OsGI* is a positive regulator of *OsMADS51*, mutation of *OsGI* may lead to reduced expression of *OsMADS51*. In our *OsFTL10* overexpression lines, the expression of *OsMADS51* was decreased. This result indicates that overexpression of *OsFTL10* might have similar effects with *OsGI* mutants on drought tolerance. Ghd7 functions in rice as a negative regulator flowering time under long days (*Xue et al., 2008*), was also found to be a negative regulator of drought tolerance in rice (*Weng et al., 2014*). As *Ghd7* acts as a suppressor of *Ehd1* under LDs, the increased expression of *Ehd1* in *OsFTL10* transgenic plants, suggests that drought tolerance conferred by OsFTL10 may be similar to *Ghd7* mutant. The link between flowering and drought

tolerance was also demonstrated by a recent work. Overexpression of *WOX13* under a drought inducible promoter Rab21 resulted in drought resistant and early flowering in rice. In these plants, drought stress and flowering related genes such as *OsDREB1A*, *OsDREB1F*, *Hd3a* and *MADS14* were up-regulated (Minh-Thu et al., 2018). Although the above analysis provide clues that drought tolerance by OsFTL10 might be related to *OsGI* and (or) *Ghd7*, through controlling the expression of *MADS51* and (or) *Ehd1*, much work including the analysis of the mutants involved in the flowering signaling pathways are needed to clarify their regulation networks. The OsFTL10 transgenic plants displayed enhanced drought stress tolerance, as revealed by the higher survival rate and better growth performance under stress conditions (Fig. 6). *Cis* elements related to drought stress and ABA response were found in the *OsFTL10* promoter (Supplemental Information 1), and qRT-PCR revealed that *OsFTL10* expression was induced by both ABA and mannitol treatments (Fig. 2). Investigation of the drought related gene expression in the *OsFTL10* transgenic lines further confirmed the drought tolerance of the plants (Fig. 6). Many transcription factors can be induced by drought stress in rice, and constitutive expression of these transcription factors have been found to improve plant tolerance to drought stress (*Shinozaki & Yamaguchi-Shinozaki, 2007*; *Nakashima et al., 2011*; *Rai & Penna, 2013*; *Janiak, Kwaśniewski & Szarejko, 2016*). We studied the expression of three transcription factors including OsDREB2A, bZIP23 and SNAC1, and found they were induced to a higher level of expression in the *OsFTL10* overexpressing line ox8. These results correlated well with the performance of the drought stress test results. DREBs/CBFs are transcription factors that can be induced by stresses. They can activate the expression of stress-inducible genes by binding to the dehydration-responsive element/C-repeat (DRE/CRT) cis-acting elements in their promoter region. OsDREB2A is a rice DREB homolog which has been found to be induced by salt and drought. The overexpression of *OsDREB2A* in rice plants showed both drought and salt stress tolerance (*Ito et al., 2006*). Abscisic acid is regarded as a drought stress hormone in plants. The transcription factor bZIP23 plays an important role in ABA signaling by binding to the promoter of PP2C49 to improve its expression. As a co-receptor, PP2C can increase ABA binding affinity toward the formation of the Receptor-ABA-PP2C complex (*Moreno-Alvero et al., 2017*). bZIP23 has been also found to improve drought tolerance in rice (*Xiang et al., 2008*). The *SNAC1* gene belongs to the stress-related NAC superfamily of transcription factors, which has been expressed in plant for improving drought tolerance (*Nakashima et al., 2011*). Although the induced expression of the above three transcription factors can be considered as evidence of OsFTL10 tolerance to drought stress, mechanistic links between OsFTL10 and transcription factors were not fully determined.

## CONCLUSIONS

In summary, results presented in this study reveal that OsFTL10 functioned similarly to Hd3a in rice by promoting floral transition, and overexpression of *OsFTL10* improved the drought stress tolerance of the plants. Further investigations on the identification of *OsFTL10* regulated target genes will be helpful for fully elucidating the linkage of flowering and drought stress in the monocot model rice.

## ACKNOWLEDGEMENTS

We thank Dr. R.A. Jefferson, CAMBIA, Canberra, Australia, for kindly providing the pCAMBIA1301 and pCAMBIA1390 vectors. We would also like to express our gratitude to Rongyu Du for technical assistance.

### Funding

This work was supported by the National Natural Science Foundation of China (No. 31571759; 31171185), the Natural Science Foundation of Guangdong Province of China (No. 2016A030313150). The funders had no role in study design, data collection and analysis, decision to publish, or preparation of the manuscript.

### Grant Disclosures

The following grant information was disclosed by the authors:
National Natural Science Foundation of China: 31571759, 31171185.
Natural Science Foundation of Guangdong Province of China: 2016A030313150.

### Competing Interests

The authors declare there are no competing interests.

### Author Contributions

- Maichun Fang and Meiru Li conceived and designed the experiments, performed the experiments, analyzed the data, prepared figures and/or tables, authored or reviewed drafts of the paper, approved the final draft.
- Zejiao Zhou performed the experiments, contributed reagents/materials/analysis tools, prepared figures and/or tables, authored or reviewed drafts of the paper, approved the final draft.
- Xusheng Zhou performed the experiments, approved the final draft.
- Huiyong Yang performed the experiments, contributed reagents/materials/analysis tools, approved the final draft.
- Hongqing Li conceived and designed the experiments, prepared figures and/or tables, authored or reviewed drafts of the paper, approved the final draft.

### Data Availability

Raw data is available in Supplemental Files.

### Supplemental Information

Supplemental information for this article can be found online at http://dx.doi.org/10.7717/peerj.6422#supplemental-information.

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
