# Peer review of "Overexpression of OsFTL10 induces early flowering and improves drought tolerance in Oryza sativa L"

_PeerJ, doi:10.7717/peerj.6422_

## Round 0.1 · original submission · Major Revisions

# I am sorry but your manuscript is not in a shape that it can be reviewed. Specifically, the important Figure 6 cannot be understood due to incomplete legends and conflicting labelling (which figure parts are A, B, a-e?; Which data points refer to drought-stressed and non-drought-stressed plants in the bottom part of the figure on gene expression and why is the Y-achsis labelled "after drought stress" for all data if only parts of the plants were drought-stressed? This makes no sense. Moreover, please provide negative controls for the BiFC experiments shown in Fig. 4A.

If you address these points in a revised version I will be happy to handle your submission and send it for peer review.

---

## Round 0.2 · Major Revisions

Your manuscript provides interesting and novel data on the function of OsFTL10. In the revised version, please pay particular attention to comments regarding the statistical analysis of your data. Please clearly indicate the number of biological replicates (i.e. independent RNA preps and independent experiments) that were analyzed for all RT-qPCR analyses. Please also indicate the identity of the error bars in your graphs (SEM or SD?).

Reviewer 1 ·

Basic reporting

1) Please explain more about functions and encoding protein of genes introduced in the manuscript. It is hard to see, for example, what “GF14c” (line 250) is even when reading the corresponding position in the manuscript. Describing functions of GI, Hd1 Ehd1 and MADS51 against flowering time (induction of flowering or suppression?) as well as their encoding proteins would help readers to understand the content more clearly.

2) The OsFTL10 overexpressors exhibits early flowering whereas expression of Ehd1, a suppressor of flowering under LDs, seems to be induced in the lines. It would be good to discuss this in the discussion part.

3) In the overexpressor Hd3a expression is induced whereas RFT1 is repressed. It would be good to discuss this observation.

4) Is the increased tolerance against drought in the OsFTL10 overexpressor lines due to increased expression of Ehd1? It would be good if you explain a bit more about the possible mechanism by which OsFTL10 overexpression mediates drought-stress tolerance in the transgenic lines.

Experimental design

Overall I am satisfied with the experimental design.

Validity of the findings

As for graphs representing expression of DREB2, bZIP23 and NAC1 (Fig. 6I, J, K), positions of asterisks are inconsistent with the figure legend. Figures look to indicate significant difference in comparison between time 0 and other times in each line and not between WT and transgenic lines. Please correct positions of asterisks in the figures comparing expression levels in transgenic lines to those in WT, and also change the corresponding sentences accurately describing the results.

Additional comments

This manuscript describes a study on a mechanism that controls the timing of flowering and its linkage to the drought stress response in rice. Authors create transgenic lines in which mRNA of a flowering-time gene OsFTL10 is overexpressed, or accumulation of its endogenous mRNA is suppressed, to examine its functions to control flowering and the drought-tolerance responses. Through a series of phenotypic analyses authors demonstrate that OsFTL10 induces flowering also increasing tolerance to drought. They also show through split-YEP and yeast two-hybrid assays that OsFTL10 physically binds to rice 14-3-3 proteins, also representing through mRNA analyses using transgenic lines that OsFTL10 affects expression of genes that control flowering. Also, they represent data that the overexpression increases some drought-stress inducible genes such as DREB2.

The experiments overall look carefully performed, and represented data also include clear messages. Therefore, I feel that this manuscript is suitable for publication to Peer J. However, I found several minor issues that would need to be addressed prior to submission.

·

Basic reporting

The manuscript reads well except for a few language issues (e.g lines 86-89; 374; quirky passive voice structures). A quick check by a language editor should be sufficient to correct those.

* The experiments with CaMV35S::Hd3a and analysis of the cis-elements (lines 349, 374) are not mentioned anywhere before the discussion. These data should be described in the M&Ms and Results sections.

* Fig. 1 Higher resolution photos of the GUS stainings are required if available. In the current version, I couldn't see the staining signal in pollen.

* Fig. 6 The meaning of the asterisks here does not seem to match the legend. Is it testing significances against a 0 time-point or against a previous time point? The significance levels between the WT, transgenic and knockdown plants should also be tested to support the claims.

* What are the whiskers on the bar graphs?

* For some statements, literature references are missing (lines 67, 337).

Experimental design

The manuscript describes a coherent set of experiments aimed to characterize the FTL10 gene - one of the previously uncharacterized FT-like paralogs in cultivated rice, and understand its role in regulating flowering time and drought tolerance. The research question is relevant and understandable but not well defined in the Introduction (lines 106-108). I.e. you did not investigate the environmental response of the florigens Hd3a and RFT1 in this manuscript.

M&Ms

* For the qRT-PCR experiments specify the sampling scheme - which tissue, how many replicates, what time of the day. For all the experiments, specify the number of replicates sampled.

* line 146 - Specify the traits.

* lines 188 - 189 Why did you use a different duration of stress for RNAi and ox lines?

Validity of the findings

The experiments reported here revealed the dual role of FTL10 in regulating flowering time and drought tolerance in rice.

* Most of the claims are supported by the presented data and the experiments include the necessary controls. If feasible, in addition to MADS14 and 15, I suggest measuring the expression of the remaining MADS genes downstream of FT - MADS18 and 34 (as described in the Introduction), which will support or expand the statement on lines 278-280. What was the evidence of the direct interaction of FTL10 and MADS15 as you formulated it here?

* By looking at the bar charts, it is challenging to assess whether the analyses and claimed differences are statistically sound without referring to the raw data (included in the supplement) and re-analysing them. Adding the individual measurement data points to all the bar charts or better re-plotting them as box plots (also with the individual data points) would solve this problem and greatly help future readers.

Discussion

* The hypotheses of why the oxFTL10 lines flowered later than the oxHd3a plants are interesting but rather speculative at this point. If you decide to leave them in the revised version, I suggest illustrating them as an additional figure describing the candidate model.

* Involvement of FTL10 in drought tolerance is an intriguing finding. I believe discussing what is known about the interplay between the flowering and drought tolerance pathways in other plant species (and it is not much) will strengthen the discussion. For example, Habte et al. (Plant, Cell & Environment, 2014) also found cis-elements of the drought-related genes in the promoters of the light signalling genes in barley and vice versa.

Additional comments

Minor issues.

* line 85 - spell out the gene names
* line 92 - spell out GA3
* line 95 - drought escape is not only an acceleration but also a delay of flowering.
* lines 103, 111 - Do you mean drought escape or indeed tolerance here?
* lines 106 - 108 Revise Here, you investigated another homolog of the rice FTLs - FTL10 but not environmental responses of Hd3a and RFT1.
* line 174 - A single solution containing the mixture of these compounds?
* lines 175 - Specify times.
* line 198 - "polypeptide protein"?
* line 198-199 Revise wording to be specific what you did. Here and throughout the text, avoid formulations such as "the function of FTL10 on drought stress was analyzed" (line 225). Write specific, e.g. "we studied the effect of drought on the expression of ..." etc.
* line 202 Specify in which tissues.
* line 207 Specify in which tissues.
* lines 235 - 239 Move to M&Ms
* line 249 How does GF14c come into play? It is not mentioned anywhere in the introduction.
* line 317 “a transcription that..” → “a transcription factor that..”
* Fig. 3. “Performance of the transgenic plants in field ..” contradicts M&M line 145 “...grown under greenhouse conditions ...”.

---

## Round 0.3 · accepted · Accept

The revised version of your manuscript addressed the editor´s and reviewers´ comments.

#